# Research Advances in Ion Exchange of Halide Perovskites

**DOI:** 10.3390/nano15050375

**Published:** 2025-02-28

**Authors:** Chao Du, Kaiwang Chen, Jiangshan Chen, Dongge Ma

**Affiliations:** Institute of Polymer Optoelectronic Materials and Devices, Guangdong Basic Research Center of Excellence for Energy & Information Polymer Materials, State Key Laboratory of Luminescent Materials and Devices, Guangdong Provincial Key Laboratory of Luminescence from Molecular Aggregates, South China University of Technology, Guangzhou 510640, China; 202220118814@mail.scut.edu.cn (C.D.); 202110183318@mail.scut.edu.cn (K.C.); msdgma@scut.edu.cn (D.M.)

**Keywords:** halide perovskites, composition engineering, ion exchange, optoelectronic characteristics, post-treatment

## Abstract

In recent years, halide perovskite materials have been extensively studied by researchers due to their excellent optoelectronic characteristics. Unlike traditional semiconductors, halide perovskites possess unique ionic crystal structures, which makes it easier to perform facile composition engineering to tailor their physical and chemical properties. Ion exchange is a popular post-treatment strategy to achieve composition engineering in perovskites, and various ion exchange processes have been used to modify the structural and functional features of prefabricated perovskites to meet the requirements of desired applications. This review summarizes the recent progress in ion exchange of halide perovskites, including mechanisms, strategies, and studies on different ion exchange. Additionally, the applications of ion-exchanged perovskites in microfluidic sensors, light-emitting diodes (LEDs), lasers, and solar cells are presented. Lastly, we briefly discuss the challenges in ion exchange of perovskites and hope that ion exchange can provide a more refined and reliable method for the preparation of high-performance perovskites.

## 1. Introduction

In recent years, significant progress has been made in the development of advanced materials for optoelectronic applications, such as solar cells, light-emitting diodes (LEDs), photodetectors, and lasers. Among these, quantum dots (0D), semiconductor nanostructures (1D), and low-dimensional systems (2D) have emerged as promising candidates. For example, quantum dots composed of materials like CdSe and PbS exhibit size-dependent optical properties and high quantum yields, making them attractive for various optoelectronic applications [1,2]. Similarly, GaAs nanowires have garnered considerable attention due to their unique optical and electronic properties [3,4]. Furthermore, research on ZnSe/ZnSeS superlattices has shown that these structures exhibit unique excitonic properties, which can be tuned through compositional and structural modifications, making them attractive for applications in optoelectronic devices [5]. Despite these advances, halide perovskites have emerged as a highly competitive alternative, offering a unique combination of superior optoelectronic properties and cost-effective fabrication processes that distinguish them from traditional materials.

Perovskites refer to a class of materials with the general formula of ABX_3_, which were first discovered by the German mineralogist Gustav Rose in 1839 to be present in calcium titanate (CaTiO_3_) compounds. In recent years, halide perovskites, as direct bandgap semiconductors, have drawn increasing attention because they possess many advantages, such as high defect tolerance, long carrier diffusion lengths, high mobilities, easily tunable bandgaps, high photoluminescence quantum yields (PLQYs), and cost-effective preparation using solution processes [6,7]. These features make halide perovskites extremely promising for applications in optoelectronic devices. During the past few years, research progress in perovskite LEDs (PeLEDs) has made significant strides, with the significant enhancement in external quantum efficiencies (EQEs) exceeding 30%, which is comparable to the traditional organic LEDs (OLEDs) and quantum dot LEDs (QDLEDs) [8,9,10,11]. Unlike the traditional oxide perovskites, the halide perovskites are composed of A-site monovalent cations such as FA^+^ (HC(NH_2_)_2_^+^), MA^+^ (CH_3_NH_3_^+^), and Cs^+^; B-site divalent cations such as Pb^2+^ and Sn^2+^; and X-site halide anions such as Cl^−^, Br^−^, and I^−^ [12,13,14,15]. In the crystal structure of halide perovskites, the B-site cation is coordinated with six X-site anions to form a [BX_6_]^4−^ octahedron, and the A-site cation is located within the cavity formed by eight corner-sharing [BX_6_]^4−^ octahedrons. So the size of various ions has a significant impact on the structure and stability of perovskite crystals according to the formula of Goldschmidt’s tolerance factor$$\tau =\frac{R_{A}+R_{B}}{\sqrt{2}\;(R_{B}+R_{X})},\;0.76\;\leq \;\tau \leq \;1.13;\;\mu =\frac{R_{B}}{R_{X}},0.44\leq \mu \leq 0.9$$
where *τ* is the tolerance factor, *μ* is the octahedral factor, and *R* is the ionic radius of the respective corresponding ion [16].

Additionally, according to the unique soft ionic crystal structure of halide perovskites, the valence band (VB) is originated from the contribution of the outer *p* orbitals of the X-site halide anions and the outer *s* orbitals of the central B-site metal cations, while the conduction band (CB) is mainly determined by the outer *p* orbitals of the B-site cations. Therefore, the bandgaps of halide perovskites can be easily modulated by changing any ionic compositions in the ABX_3_ lattice, especially the B-site divalent metal cations and X-site halide anions [17]. Moreover, the structural stability of halide perovskites is closely related to the A-site cations due to their influence on the [BX_6_]^4−^ octahedron framework [16]. Therefore, the optical and electrical properties, as well as the structural stability of halide perovskites, can be precisely tuned by composition engineering within the permissible range of *τ* and *μ* to meet the requirements of practical applications.

The structural and optoelectronic properties of perovskite materials are significantly influenced by their preparation methods. Common synthesis approaches encompass solution-based techniques, such as spin-coating [11,18], and vapor-based methods, such as physical vapor deposition (PVD) [19]. These methods typically involve the formation of perovskite films through the reaction of metal halides with organic ammonium halides in appropriate solvents or under controlled conditions. For perovskite nanocrystals, several synthesis methods have been developed to achieve control over their size, shape, and composition. Among these, the hot-injection method is extensively utilized [20]. This method entails the rapid introduction of metal halide precursors into a heated solution containing organic ligands, facilitating the formation of high-quality nanocrystals characterized by narrow size distributions and high photoluminescence quantum yields. Another method is ligand-assisted reprecipitation, which uses specific ligands to regulate the nucleation and growth of nanocrystals at ambient temperature [12,21]. Additionally, halide anion exchange methods have been employed to tune the composition and optical properties of perovskite nanocrystals by exchanging halide ions in the crystal lattice [22]. Various methods for preparing perovskite materials also have different characteristics. Solution-based methods are cost-effective and scalable but require optimization to achieve uniform film coverage and high crystallinity. Vapor-based methods provide better control over film uniformity and thickness but are more complex and expensive. The synthesis of perovskite nanocrystals through hot-injection and ligand-assisted reprecipitation enables precise control over their morphology and optical properties, enhancing their performance in optoelectronic devices.

Among the various methods of composition regulation for halide perovskites, ion exchange strategy is favored by researchers due to its simplicity and significant effectiveness. Ion exchange is a post-treatment method that occurs through ion diffusion, driven by ion concentration differences between the pre-prepared perovskites and their surrounding medium. The inherent ionic lattice and soft crystal structure of halide perovskites enable ion exchange to be a useful method for tailoring their physical and chemical properties [22,23,24,25]. In this review, we introduce the mechanism of ion exchange in halide perovskites and summarize the strategies of ion exchange under different states of precursors. Subsequently, we outline the current status of research on anion exchange and cation exchange, respectively. Finally, we briefly introduce the applications of ion-exchanged perovskites. Our aim is to provide a guide to rationally conduct ion exchange for the structural design and composition optimization of halide perovskites.

## 2. Mechanism and Strategy of Ion Exchange in Halide Perovskites

### 2.1. Ion Exchange Mechanism

Ion exchange of perovskites is a method of introducing other ions from the surroundings into perovskite crystals, thereby partially or completely replacing the ions originally present in the perovskites. Therefore, the process can be roughly divided into two parts: diffusion and exchange.

The mass transfer process in ion exchange is initially characterized by the interactive diffusion between external ions and the parent compound. Diffusion is the process by which atoms, ions, or molecules migrate within a substance due to thermal motion, and the kinetics of this process are typically described by Fick’s laws of diffusion [26]. During the ion exchange process in perovskite materials, after introducing ion exchange precursors, mutual diffusion of ions occurs between the exchange precursors and the original perovskites. This process is a migration behavior caused by the thermal motion of ions, so the diffusion rate is strongly dependent on temperature and ion concentration. Actually, the ion migration is closely related to the ionic defects in halide perovskites, particularly the vacancies. Those defects can lower the activation energy for ion migration and provide easier pathways for ion exchange [27]. Various vacancy-mediated diffusions occur within the perovskite crystal lattice [28]. In materials with the ABX_3_ perovskite structure, vacancy-mediated diffusion is the most common process (Figure 1) [29]. This includes (i) X^−^ migration along an octahedron edge; (ii) B^2+^ migration along the diagonal (<110> directions) of the cubic unit cell; and (iii) A^+^ migration into a neighboring vacant A-site cage. Taking MAPbI_3_ for example, the intrinsic concentrations of I^–^, Pb^2+^, and MA^+^ vacancies exceed 0.4% at room temperature, which makes vacancy diffusion dominant in MAPbI_3_ crystals [30]. Vacancy diffusion is typically a jump transfer process, which relies on thermodynamic fluctuations within the lattice. The number of jumps per second is called the jump rate and can be expressed by the following formula:$$\nu =\nu_{0}e^{-\varepsilon /k_{B}T}$$
where $\nu_{0}$ is the lattice vibration frequency, *ε* is the atomic transfer barrier, $k_{B}$ is the Boltzmann constant, and *T* is the temperature [31].

“Exchange” in the ion exchange of perovskites mainly refers to the process where new ions enter the perovskite crystal while the corresponding original ions migrate out. As shown in Figure 2, in the crystal structure of ABX_3_ perovskites, the X-site halide anions are located at the vertices of the [BX_6_]^4−^ octahedron and have the smallest activation energy, making them easier to undergo ion exchange [32]. In contrast, the A-site cations are located at the empty sites surrounded by eight corner-sharing [BX_6_]^4−^ octahedrons. The migration path of these cations involves passing through either a single crystal face or a ‘bottleneck’ composed of four X-site anions, leading to higher activation energy [33]. Additionally, the A-site cations usually have a larger volume, especially in organic–inorganic hybrid perovskites, so their migration can significantly influence the stability of perovskite crystals. On the other hand, the B-site cations located at the center of the [BX_6_]^4−^ octahedron exhibit the highest migration energy barrier. For example, in the crystal structure of MAPbI_3_ perovskite, the activation energies of the A-, B-, and X-sites are 0.84, 2.31, and 0.58 eV [29], respectively. Therefore, ion exchange is easier to occur at the X-site anions than the A-site and B-site cations.

### 2.2. Strategy of Ion Exchange

Due to the unique crystal structure of perovskites with [BX_6_]^4−^ octahedrons sharing common vertices, they have a larger lattice spacing and higher tolerance for crystal defects compared to common edge- or face-shared octahedral crystal structures [34]. Therefore, this also makes ion diffusion and migration within perovskite crystals relatively easy, leading to relatively facile ion exchange in perovskites. Here, we roughly classify perovskite ion exchange into solid-state exchange, liquid-state exchange, and gas-state exchange based on the state of the ion exchange precursor (Figure 3).

Solid-state exchange refers to ion exchange reactions that occur between the ion exchange precursors and perovskite compounds under solid-state conditions. This type of reaction typically involves contacting ion exchange precursor with the perovskite surface. Subsequently, ion exchange occurs at the contacting surface and gradually diffuses into the deep layers of the perovskite until it reaches equilibrium [35,36]. However, due to diffusion limitations, solid-state exchange often struggles to achieve complete ion exchange.

Liquid-state exchange refers to ion exchange reactions that occur between the ion exchange precursors dissolved in specific liquid solvents and perovskite-dispersed solutions. Liquid-state exchange reactions typically occur quickly, reaching equilibrium in a matter of seconds. Additionally, liquid-state exchange allows for complete ion exchange of all three elements in the perovskite [37,38], making this exchange strategy widely used.

Gas-state exchange refers to ion exchange reactions between the ion exchange precursors carried by gas and perovskites. This strategy allows for control of the ion exchange rate by manipulating the evaporation rate of the ion exchange precursor [39,40]. However, this kind of ion exchange imposes more stringent requirements for the ion exchange precursors.

## 3. Anion Exchange of Perovskites

### 3.1. Bandgap Regulation

The X-sites in perovskites are occupied by halogens, and the bandgaps of perovskites are varied with the different halogens. Generally, the bandgaps of pure chloride perovskites, pure bromide perovskites, and pure iodide perovskites are located in the ultraviolet, green, and infrared regions, respectively. By mixing different halides, the emission peaks of perovskites can be adjusted continuously within the entire visible light range (390 nm–780 nm) [41,42]. Anionic exchange, as a post-treatment method, is often used to regulate the bandgaps of perovskites. Currently, the commonly used anionic exchange method is to treat the pre-synthesized perovskite with halide solution to achieve halide exchange. Common halide sources include lead halide (PbX_2_) [43,44], zinc halide (ZnX_2_) [45], cesium halide (CsX) [46], Grignard reagents (RMgX) [43], lithium halide (LiX) [41], and alkylammonium halide (RAmX) [23]. By adjusting the concentration of different halides, the emission peak of perovskite can be continuously adjusted within the visible light range. Xu et al. chose trioctylphosphine (TOP) to replace traditional oleic acid and oleylamine as surface ligands and then CsPbI_3_ perovskite QDs with high PLQY and satisfying stability are synthesized successfully. Subsequently, hydrobromic acid and hydrochloric acid were selected as the sources of halogen ions. A facile anion exchange process was realized through an oil/water biphasic immiscible system, tuning the emission spectra of CsPbX_3_ quantum dots from 423.0 nm to 670.5 nm [47]. Huang et al. proposed a novel surface cation cross-linking (SCCL) strategy to deal with the original CsPbBr_3_ PNCs for the cross-locking of surface uncoordinated Pb [48]. They employed a dendritic ligand of pentaerythritol tetrakis(3-mercaptopropionate) (PETMP) to form multiple coordination bonds with the surface Pb cations through its four end-thiolates and four side-carboxylates. After anionic exchange by oleylamine (OAm) with the chloride and iodide salts of ZnCl_2_ and ZnI_2_, the resulting perovskite NCs exhibited high PLQYs of over 90% at the full-color spectra from 450 nm to 720 nm, with the optimized RGB three types of NCs all reaching up to 97% PLQY, demonstrating that the halide content can be adjusted through anion exchange to achieve “on-demand” synthesis of perovskite NCs with desired colors. Mishra et al. found that using HX (X = Cl and Br) as precursors for anionic exchange can maximize the photoluminescence (PL) characteristics of CsPbBr_3_ NCs, and these hydrogen halide–based anionic exchanges were found to be completely reversible [49]. They conducted anionic exchange between HCl and CsPbBr_3_ NCs to change the emission peak from 512 nm to 428 nm. Subsequently, they also conducted anionic exchange between the perovskite NCs with an emission peak of 428 nm and HBr to achieve spectral reversal (Figure 4a) [49]. In addition, Wang et al. reported an in situ halide exchange approach by using multi-functional organic halide salts [50]. They used tetraphenylphosphonium chloride (TPPC) as the chloride source to perform an in situ halide exchange reaction, shifting the emission peak of CsPb(Br_0.65_Cl_0.35_)_3_ films from 481 nm (sky blue) to 455 nm (deep blue) (see Figure 4b). Furthermore, they treated the CsPb(Br_0.65_Cl_0.35_)_3_ film with a solution of tetraphenylphosphonium bromide (TPPB), and showed the redshift of PL spectra due to the substitution of chloride ions by bromide ions, which indicates that this in situ halide exchange is bidirectional. Manna et al. reported the exchange reaction directly from CsPbCl_3_ to CsPbI_3_ in nanocrystals (NCs), and revealed the reaction mechanism [51]. They used PbI_2_ dissolved in a mixture of OAm and oleic acid (OA) as a halide source to perform the halide exchange reactions, shifting the emission peak from blue to red. It was found that the reaction initially formed iodide-doped CsPbCl_3_ NCs, which were enveloped by a monolayer shell of CsI. These NCs leaped over the miscibility gap between CsPbCl_3_ and CsPbI_3_ by briefly transitioning to short-lived and nonrecoverable CsPb(Cl_x_I_1−x_)_3_ NCs, which quickly expelled the excess chloride and turned into the chloride-doped CsPbI_3_ nanocrystals [51].

Generally, the speed of liquid-state anionic exchange reactions is very fast, and the entire reaction process can be completed within minutes or even seconds, which makes it difficult to precisely control the liquid-state exchange reaction. To avoid this limitation, researchers have attempted to use gas-state exchange methods to perform the anion exchange of perovskites. This strategy can control the exchange speed by regulating the evaporation rate of the halide source [39,52,53,54,55,56]. For example, Ye et al. placed the CsPbBr_3_ films in an atmosphere containing diphenylphosphinic chloride (DPPOCl) to undergo anionic exchange. This method prolonged the reaction time to 2 h and ultimately yielded a pure blue-emitting perovskite film with an emission peak of 471 nm [52]. In this reaction, DPPOCl reacted with trace amounts of water to produce DPPOOH while releasing free Cl^−^. Subsequently, the separated Cl^−^ was introduced into the perovskite lattice and partially replaced Br^−^ [52]. In addition to preparing blue-emitting perovskites, green- and red-emitting perovskites can also be prepared using this method. Li et al. controlled the heating process of MABr powder to regulate its vapor generation rate and slowly conducted anionic exchange reactions with MAPbI_3_ films to adjust the emission peak from 771.1 nm to 540.5 nm [53].

Therefore, anionic exchange provides a simple yet reliable way to adjust the emission peak of perovskites within a large range, which offers a way to synthesize multi-color light-emitting perovskites.

### 3.2. Factors Affecting Anion Exchange

Anion exchange has the advantages of simple operation and significant effectiveness in regulating the bandgaps of perovskites, making it highly favored by researchers [57,58,59]. However, the current understanding of the underlying mechanism of anion exchange using various halide sources is limited and remains to be explored [49,60]. According to coordination field theory, the instability of halide ions is owing to the ease of ion bond breakage and their ability to participate in anion exchange reactions. In 2019, Chen et al. discovered that the halides with higher bond energies are unfavorable for ion exchange [32]. They used a series of divalent metal halide salts MX_2_ (M = Ca, Sr, Ba, Zn, Sn, Mn, Cd, Cu, Pb, Ba) to induce halide exchange in the CsPbBr_3_ NCs system at room temperature (Figure 5). Most metal halide salts can undergo anion exchange reactions with CsPbBr_3_ NCs to achieve continuous spectral shifting. However, salts with higher bond energy such as NaCl (420 kJ/mol), BaCl_2_ (436 kJ/mol), and KCl (441 kJ/mol) will not undergo anion exchange reactions, regardless of their concentration, even at high temperatures. Although anion exchange reactions have significant tolerance for differences in bond strength between metal halide salts and CsPbBr_3_ crystals, there is an upper limit. Additionally, Abolhasani et al. discovered that salts with higher electronegativity differences between metal cations and halide anions are more soluble in mixed solvents (toluene combined with OAm), thereby facilitating halide anion migration to perovskite NCs during anion exchange reactions [61].

On the other hand, it is also important to understand the role of the cations released after the bond cleavage of the halide sources. Mishra et al. conducted anion exchange using NaCl, KCl, HCl, NH_4_Cl, HONH_3_Cl, and MgCl_2_ as halide sources to investigate the effect of different cations on halide ion migration [49]. They ultimately found that hydrogen halides had the maximum ion migration rate and fastest reaction speed in halide ion migration. Oh et al. discovered that NH_4_^+^ and Na^+^ can respectively promote Cl^−^ and I^−^ exchange with CsPbBr_3_ NCs, favoring the formation of CsPbCl_3_ and CsPbI_3_ NCs [62]. These phenomena can be explained by hard–soft acid-base theory. Compared to NH_4_^+^, Na^+^ is a relatively hard cation with a smaller ion radius. I^−^ is known as a relatively soft species in halides; therefore, its binding energy with Na^+^ is weaker. Using NaI as the exchange precursor, I^−^ can effectively react with CsPbBr_3_ NCs to carry out the anion exchange. Similarly, NH_4_^+^ soft cations have weaker binding energy with Cl^−^ hard anions. Considering that the bonds in NH_4_Cl are easily broken, they can easily donate Cl^−^ to CsPbBr_3_ NCs. In addition, NH_4_^+^ and Na^+^ appear to be more effective in passivating the surface as their ion radii are smaller. Therefore, NH_4_Cl and NaI can promote anion exchange and improve the stability of CsPbX_3_ NCs after ion exchange [62].

In addition, there is some debate regarding whether the crystal structure of perovskite changes during anion substitution. For example, Zeng et al. carried out the experiment of halide exchange between MAPbI_3_ films and CsPbBr_3_ quantum dots (QDs), and proposed that the I-Br exchange was driven by strong halide bond interactions at the MAPbI_3_/CsPbBr_3_ interface [36]. Due to interface hopping, gradient energy levels were formed. According to atomic force microscope and high-magnification scanning electron microscope results, the morphology of perovskite films and CsPbBr_3_ QDs was well preserved. On this issue, Tian et al. proposed a lattice reconstruction model, which contains three steps: disintegration, anion exchange, and lattice reconstruction. According to this model, perovskite NCs collapse prior to anion exchange [63]. There are still many controversies about the mechanism of anion exchange that require further exploration.

## 4. Cation Exchange of Perovskites

### 4.1. A-Site Cation Exchange

Currently, the reports on A-site cation exchange involve organic–inorganic hybrid perovskites (with organic cations at the A-site) [64,65,66]. Organic–inorganic hybrid perovskites exhibit unique electronic and optical properties that are distinct from traditional inorganic perovskites, leading to additional functionalities through ion exchange reactions at the A-site [67]. For example, the A-site cation exchange reactions can regulate the bandgap, environmental stability, and grain size of organic–inorganic hybrid perovskites through introducing organic cationic species such as FA^+^ and GA^+^ (CH_6_N_3_^+^) [68,69,70,71]. Therefore, ion exchange reactions at the A-site enhance the performance of organic–inorganic hybrid perovskites, greatly diversifying the types of perovskite materials. Chen et al. added formamidine acetate (FAAc) to a toluene solution of MAPbBr_3_ NCs to carry out the exchange reaction between FA^+^ and MA^+^ [72]. The low solubility of FAAc in toluene led to slow ion exchange reaction while maintaining particle stability. It was found that the PL peak of MAPbBr_3_ NCs shifted from 515 nm to 531 nm and did not show further red shifts with prolonged reaction time, indicating that the A-site exchange reaction reached equilibrium. This work demonstrates that MA^+^ ions can be fully replaced by FA^+^ ions to generate FAPbX_3_ NCs while maintaining the initial size, shape, and crystal structure of original MAPbX_3_ NCs [72]. Similarly, Snaith et al. soaked MAPbI_3_ or FAPbI_3_ in solutions of formamidine iodide (FAI) or methylammonium iodide (MAI) to exchange ions between different perovskite materials and finely tune the bandgaps between 1.57 and 1.48 eV [73].

In addition to the exchange between organic cations and organic cations, A-site ion exchange can also occur between inorganic cations and organic cations. Luther et al. combined the colloidal solutions of CsPbI_3_ and FAPbI_3_ NCs with a suitable ratio and moderate heating (activation energy of about 0.65 eV) to allow the A-site cation exchange, resulting in Cs_1−x_FA_x_PbI_3_ alloy NCs that retained their crystal framework [74]. This process achieved emission tunable over the range of 650–800 nm solely by varying the A-site cation composition (Figure 6). They used the resultant Cs_1−x_FA_x_PbI_3_ NCs to fabricate solar cell devices that showed improved open circuit voltage (V_OC_) (compared to the similar devices with the bulk grain films as the active layers) and low hysteresis with a power conversion efficiency (PCE) of around 10% [74]. Son et al. proposed the thermal annealing–induced solid-state cation exchange at the surface of CsPbI_3_/FAPbI_3_ bilayer to develop a graded heterojunction for the fabrication of efficient and stable solar cells [75].

Introducing larger organic cationic groups into the A-sites of perovskites typically results in the change of their size and dimensionality [76,77]. Balakrishnan et al. prepared the MAPbBr_3_ NCs by a room temperature synthesis method, and then gradually added phenethylamine cations (PEA^+^) into the perovskite NCs solution for ion exchange [78]. As the content of PEA^+^ was increased, the PL emission from the perovskite solution under UV illumination changed from green to purple, which was attributed to the structural transformation from three-dimensional (3D) bulky crystals to two-dimensional (2D) layered crystals. It was found that the ion exchange reactions can be achieved between the large-sized cations of PEA^+^ and the small-sized cations of MA^+^.

### 4.2. B-Site Cation Exchange

In halide perovskites, the B-site cation is located at the center of an octahedron with the structure of [BX_6_]^4−^, so the B-site has a larger coordination number compared to the A- and X-sites [79]. It is generally believed that the B-site cation exchange needs to first open the rigid [BX_6_]^4−^ octahedron through anion exchange [80,81]. Chen et al. reported the rapid cation exchange of Mn-Pb at room temperature with the help of dynamic halide exchange (Figure 7a) [80]. They conducted the experiments using different soluble Mn^2+^ salts (Mn(Ac)_2_, C_15_H_21_MnO_6_, and Mn(OA)_2_) and Cl^−^ salts (NH_4_Cl, ZnCl_2_, GdCl_3_, and SnCl_4_) to provide Mn^2+^ and Cl^−^ ions for ion exchange with CsPbX_3_ (X = Cl and Br) NCs. They found that the cation exchange between Mn^2+^ and Pb^2+^ highly depended on the Cl^−^ concentration in the solution of exchange precursors, rather than the Mn^2+^ concentration. By opening the rigid [PbX_6_]^4−^ octahedron through dynamic Cl-Cl anion exchange, Mn^2+^ completed rapid exchange with Pb^2+^ within seconds. When mixing CsPbBr_3_ with MnBr_2_, the Mn-to-Pb cation exchange cannot proceed, because the energy difference between the Mn-Br bond and the Pb-Br bond is large, making Mn tend to combine with Br and stay in the solution rather than enter into the crystal lattice of CsPbBr_3_. Therefore, it is necessary to use other Mn compounds instead of MnBr_2_ to provide the Mn source and add a Cl source to trigger the Cl-to-Br anion exchange, which temporarily opens the [PbBr_6_]^4−^ octahedron and induces the cation exchange of Pb by Mn in the CsPbBr_3_ host. Similarly, Yang et al. also found that Mn^2+^ doping could be achieved through the B-site cation exchange with the assistant of halide ions, which was confirmed by the characteristic dual-emission peaks from the Mn-doped CsPbX_3_ NCs [82]. By contrast, the emission of Mn^2+^ ions was not observed in the absence of halide ions. Chen et al. investigated the solid-state B-site exchange between the MnCl_2_·4H_2_O solid and CsPbCl_3_ NCs powder, and demonstrated the doping of Mn^2+^ into CsPbCl_3_ NCs at room temperature without additional pressure and solvent [83]. During the solid-state exchange between Mn^2+^ and Pb^2+^, it was found that the ligands had a certain impact on the ion exchange reaction. Specifically, excessive OA hindered the doping of Mn^2+^, while excessive OAm promoted the doping of Mn^2+^.

Additionally, other divalent metal cations such as Zn^2+^ [84], Ni^2+^ [85,86], and Sn^2+^ [79,87] have also been commonly used for the B-site cation exchange in the Pb^2+^-based perovskites. Yang et al. employed SnX_2_ (X = I, Cl) as the Sn^2+^ precursors to exchange cations with CsPbBr_3_ NCs [87]. They found that the B-site cation exchange reaction only occurred when the halides of the SnX_2_ precursor and CsPbX_3_ NCs were not identical. When SnBr_2_ was used as the Sn^2+^ precursor to react with CsPbBr_3_ NCs, no exchange between Sn^2+^ and Pb^2+^ occurred within a short period of time. These results clearly indicated that the cation exchange between Sn^2+^ and Pb^2+^ required the help of anion exchange (anion exchange breaks the Pb-Br bonds in the [PbBr_6_]^4−^ octahedron, providing the driving force for cation exchange between host Pb^2+^ and guest Sn^2+^). Lifshitz et al. found that the cation exchange driven by anion exchange can introduce Ni^2+^ into the B-site of CsPbBr_3_ and CsPb(Br/Cl)_3_ NCs under room temperature [85]. They demonstrated that Ni^2+^ cations were uniformly distributed in the host lattices when controlling the Ni^2+^ concentration at around 1% to 12% during the exchange. As a result, the Ni^2+^-doped perovskite NCs exhibited excellent chemical and photochemical stability, which is crucial for applications in solar cells. Vanmaekelbergh et al. used cation exchange to partially replace Pb^2+^ with the divalent cations of Sn^2+^, Cd^2+^, and Zn^2+^ to obtain the alloyed NCs of CsPb_1−x_M_x_Br_3_ (M^2+^ = Sn^2+^, Cd^2+^, and Zn^2+^), as shown in Figure 7b [81]. The exchange of Pb^2+^ with M^2+^ resulted in a blue shift of the absorption and emission spectra, retaining the high PLQYs (>50%) and narrow PL line width (80 meV) of the parent perovskite NCs. The blue shift was attributed to the contraction of the cubic perovskite unit cell, which shortened the Pb-Br bonds and increased the interaction between Pb and Br orbitals.

## 5. Applications of Ion-Exchanged Perovskites

The direct application of ion exchange in perovskites is the development of sensors for assaying the halide content in the environment by detecting the change in optical properties of perovskites. For example, Chen et al. deposited CsPbCl_3_, CsPbBr_3_, and CsPbBr_0.5_I_2.5_ NCs in the microchannels and designed a six-channel paper microchip containing these perovskite NCs for haloalkane assays [88]. This microfluidic sensor can monitor the fluorescence color changes in the CsPbCl_3_ and CsPbBr_0.5_I_2.5_ channels to qualitatively detect CH_2_Br_2_, CH_2_Cl_2_, or CH_2_Br_2_/CH_2_Cl_2_ mixture (Figure 8). Although this type of perovskite-based sensors using anion exchange reactions has some advantages such as fast response and convenient operation, current research is still limited to qualitative analysis. Therefore, future work would focus on designing flexible microfluidic sensors applicable for quantitative analysis as well as visual and instrumental readouts.

As is well known, halide perovskites possess advantageous optical properties such as wide color gamut, narrow emission bandwidth, and high PLQYs, making them extremely attractive for display applications [89,90]. Through ion exchange reactions, it is possible to quickly and cost-effectively synthesize perovskite materials with the desired emission, which is of great significance in various display devices [43]. For example, Kido et al. performed anion exchange using long alkyl–based oleylammonium iodide (OAM-I) and aryl-based aniline hydoroiodide (An-HI) with green CsPbBr_3_ NCs to prepare red CsPb(Br/I)_3_ NCs for efficient PeLEDs (maximum EQE exceeding 20%) with the emission peak at 649 nm that meet the requirement of Rec. 2020 [24]. Anion exchange can also be used to prepare the perovskite films for blue PeLEDs, addressing the issues caused by limited solubility of metal chlorides such as CsCl and PbCl_2_ [91,92,93]. Kim et al. immersed the pre-prepared thin films of CsPbBr_3_ in a mixed solution of chloroform (CF) and tributylphosphine (TBP) to perform anion exchange and obtained CsPb(Br/Cl)_3_ for spectral stable blue PeLEDs [93]. Our group demonstrated a method to prepare mixed halide perovskite NCs with stable pure red emission and high PLQY through simultaneous halide exchange and ligand exchange [22]. They converted CsPbBr_3_ NCs synthesized at room temperature into CsPbBr_x_I_3−x_ NCs by introducing ZnI_2_ for anion exchange (Figure 9a). ZnI_2_ not only provided iodine ions but also acted as an inorganic ligand to passivate surface defects and prevent ion migration, suppressing non-radiative losses and halide segregation. By regulating the amount of ZnI_2_, pure red CsPbBr_x_I_3−x_ NCs with organic–inorganic hybrid ligands achieved near-unity PLQY with a stable emission peak at 640 nm. The CsPbBr_x_I_3−x_ NCs can be combined with the parent green CsPbBr_3_ NCs and commercial blue LED chips to construct white LEDs with high color gamut (Figure 9b).

Ion exchange can also be used as an effective method to modify the perovskite materials for application in high-performance lasers. The most attractive advantage of all-inorganic cesium lead halide perovskites is their optical gain over broad spectral ranges, making them highly suitable for use in tunable lasers. The desired halide-variable perovskites can be achieved through anion exchange reaction. Zhang et al. demonstrated a vapor-solid anion exchange approach, which is suitable for precise control of the reaction process [94]. In their study, they placed the CsPbBr_3_/CsPbI_3_ films and a specific amount of PbCl_2_/PbBr_2_ powder in a sealed petri dish. By carefully controlling the heating temperature, PbCl_2_/PbBr_2_ powder partially evaporated to release vapor, enabling a vapor/solid anion exchange reaction. By adjusting the reaction time, CsPbCl_x_Br_3−x_/CsPbI_3−x_Br_x_ films with varying Cl/Br contents were obtained (Figure 10). When the reaction temperature was maintained at 90 °C, the amplified spontaneous emission (ASE) peak was linearly tuned from 544 nm to 472 nm. Compared with the CsPbClBr_2_ films obtained through the liquid-phase anion exchange method, the perovskite films fabricated using vapor/solid anion exchange technology demonstrated superior film quality and enhanced ASE performance.

In addition to being used as emitters, halide perovskites can also serve as active materials in solar cells owning to their high light absorption coefficients and carrier mobilities. For the fabrication of solar cells, it is best to employ the semiconductor materials that absorb the entire visible light and near-infrared (NIR) light [95]. Ion exchange can be used to produce mixed halide perovskite films with widely adjustable absorption spectra, low exciton binding energy, and long charge carrier diffusion length, making it popular in fabricating photovoltaic devices [73,74,96,97]. For example, Buecheler et al. prepared the perovskite solar cells with gradient-composition absorption layers through ion exchange [97]. They prepared gradient-composition absorption layers by spin coating organic bromide (MABr or FABr) solution onto the primary perovskite films (MAPbI_3_ or FAPbI_3_) for halide ion exchange. The partial ion-exchange reaction includes three different stages: (i) the starting absorber MAPbI_3_ was prepared by hybrid thermal evaporation/spin coating method; (ii) the as-prepared absorber was subjected to a post-deposition treatment by spin coating MABr solution in isopropanol to induce halide ion-exchange at room temperature; and (iii) the MABr-treated film was thermally annealed under a chlorobenzene vapor atmosphere to facilitate ions diffusion and redistribution, and to promote the crystal growth (Figure 11a). The gradient-composition perovskite absorption layers are believed to have better charge carrier mobility, which will enhance the transport of photogenerated holes and electrons. Ultimately, they achieved an improved V_OC_ of 1.116 V and PCE of 16.8% in the transparent perovskite solar cells with graded MAPbI_3−x_Br_x_ absorber layer (Figure 11b). The partial ion-exchange approach offers a viable strategy for tailoring the composition and morphology of mixed-perovskite absorbers, which are not readily accessible through other methods.

## 6. Conclusions and Outlook

This paper provides a brief overview of research on ion exchange of halide perovskites, including exchange mechanisms, exchange strategies, and different ionic exchange reactions. Due to the soft ionic crystal structure of perovskites, ion exchange can be used to easily change the composition of perovskite crystals to achieve the “on-demand” post-treatment. Moreover, based on the unique band structure of halide perovskites, changes in composition can easily tune their semiconductor properties for optoelectronic applications. Typically, the X-site anion exchange is used to change the optical properties of perovskites, and the B-site cation exchange can not only change the optical properties, but also improve crystal stability and reduce the content of toxic Pb. The exchange of A-site cations also can be used to improve the structural stability of perovskites due to their unique position in the perovskite framework. Overall, ion exchange provides a simple and feasible method for the synthesis of perovskites with desired structural and functional characteristics.

Despite the advantages of ion exchange in optimizing perovskite properties, several challenges remain. Firstly, the stability of perovskite materials remains a pivotal factor impeding their extensive application. Environmental factors such as moisture, oxygen, and ultraviolet radiation are known to induce structural degradation in perovskite materials. Particularly in applications like solar cells and LEDs, perovskite materials are susceptible to significant efficiency reduction under ambient conditions due to their instability. On this basis, ion exchange processes in perovskites often introduce stability issues that limit their practical application. For instance, the presence of volatile organic cations, such as MA^+^ and FA^+^, can lead to thermal and chemical degradation, especially under prolonged exposure to heat, light, or oxygen. In addition, the rate and uniformity of ion exchange are influenced by several factors, including the type of surface ligand, temperature, and solution environment. These factors can introduce variability in the ion exchange process, thereby reducing repeatability.

Therefore, to further expand ion exchange strategies for preparing high-performance perovskites that meet practical application requirements, several current issues need to be addressed:

(1) Developing a systematic ion exchange mechanism. Currently, the conclusions on the ion exchange mechanism of perovskites still remain at observing and conjecturing experimental results, and different research teams have various understandings on the exchange mechanism.

(2) Detailed research on exchange precursors. Most of the exchange precursors used for ion exchange in perovskites contain counterions that may not participate in the exchange reactions, which can indirectly affect the ion exchange process. These additional counterions may accelerate or hinder the main ion exchange process, and some counterions may even induce the decomposition of perovskite crystals. Detailed understanding of the characteristics of different precursors can help further accurately control the composition and structure changes of halide perovskites during the ion exchange reactions.

It is believed that with further development in ion exchange of perovskites, we will be able to clearly identify the required ion exchange precursors, reaction conditions and mechanisms, and other influencing factors to prepare “on-demand” perovskite materials for high-performance optoelectronic devices.

## Figures and Tables

**Figure 1 nanomaterials-15-00375-f001:**
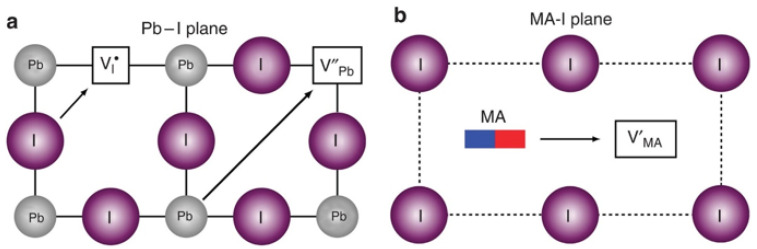
Schematic diagrams of three ion migration mechanisms involving vacancy jumps between adjacent sites: (**a**) I^−^ migration along octahedral edges; Pb^2+^ migration along diagonal directions <110>; (**b**) MA^+^ migration along the normal direction of a single-cell surface composed of four iodide ions to the adjacent A-site vacancy [29].

**Figure 2 nanomaterials-15-00375-f002:**
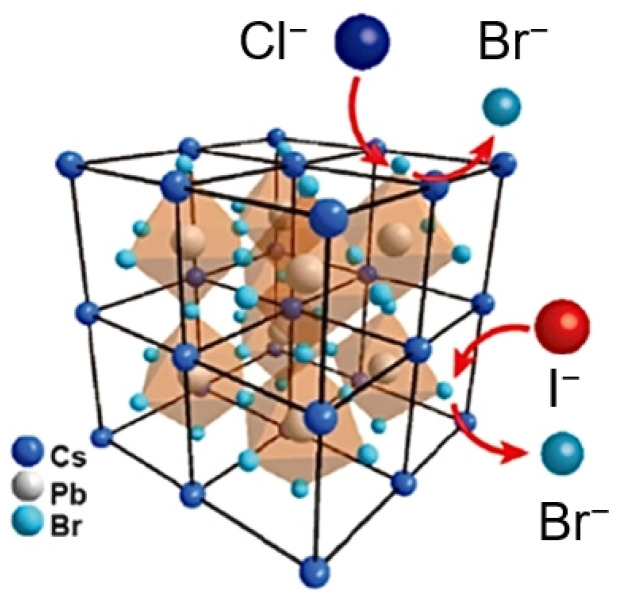
Reaction scheme of anion exchange for CsPbX_3_ [32].

**Figure 3 nanomaterials-15-00375-f003:**
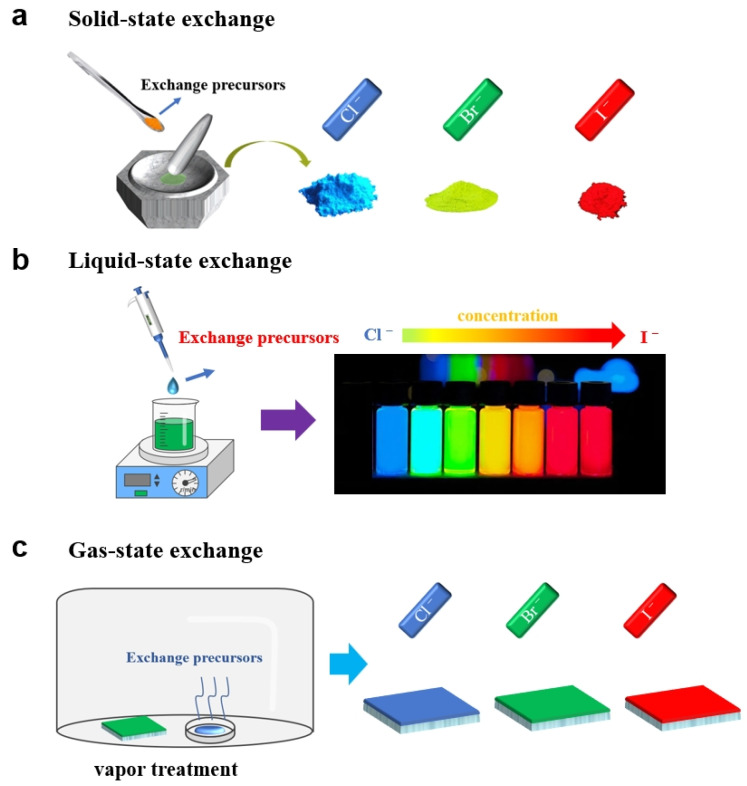
Schematic diagram of (**a**) solid, (**b**) liquid, and (**c**) gas state ion exchange processes.

**Figure 4 nanomaterials-15-00375-f004:**
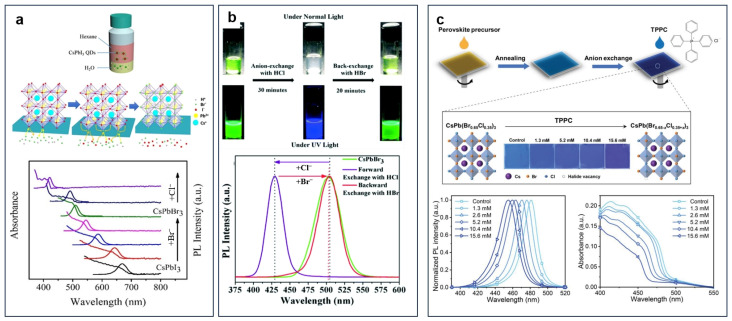
(**a**) Anion exchange of CsPbX_3_ NCs by using hydrogen halide (HX = Cl, Br, I) [47]; (**b**) anion exchange of CsPbBr_3_ NCs with HCl and then back-exchange with HBr [49]; (**c**) in situ anion exchange in CsPb(Br_0.65_Cl_0.35_)_3_ films with organic halide salts [50].

**Figure 5 nanomaterials-15-00375-f005:**
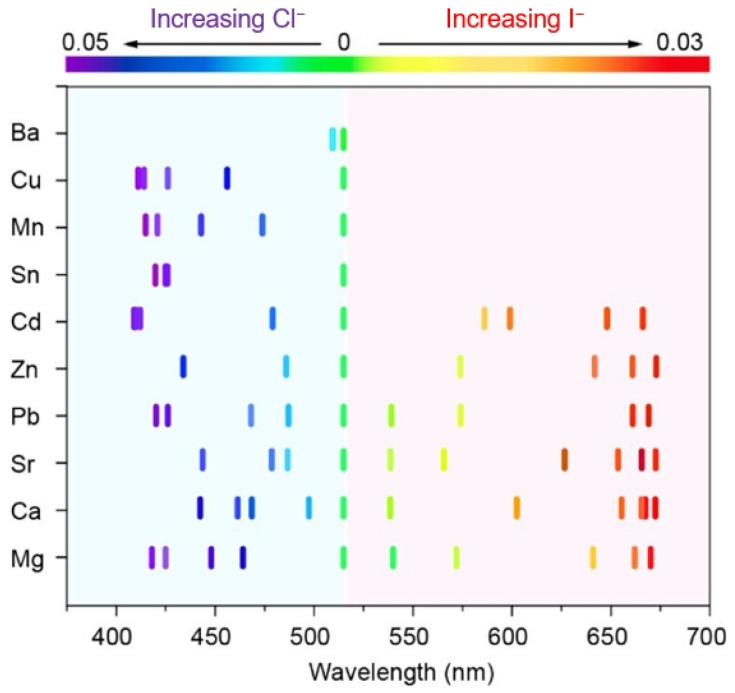
PL peaks of CsPbX_3_ NCs using various amounts of metal halides as precursors for the anion exchange of CsPbBr_3_ NCs [32].

**Figure 6 nanomaterials-15-00375-f006:**
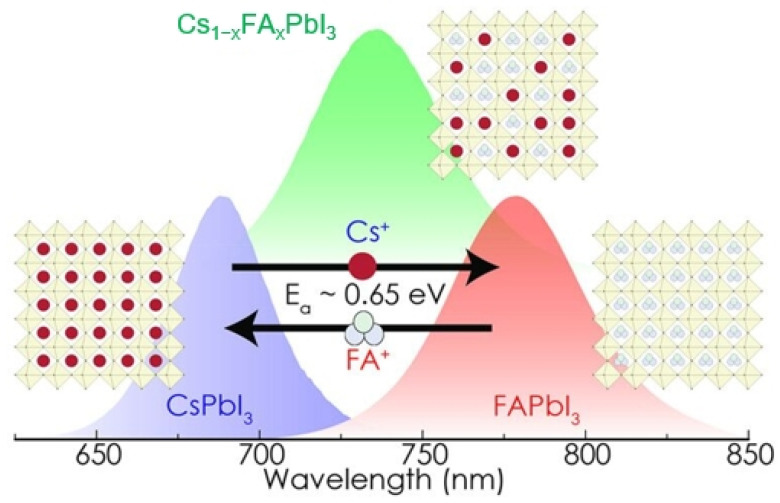
Schematic diagram of A-site cation exchange between CsPbI_3_ and FAPbI_3_ NCs [74].

**Figure 7 nanomaterials-15-00375-f007:**
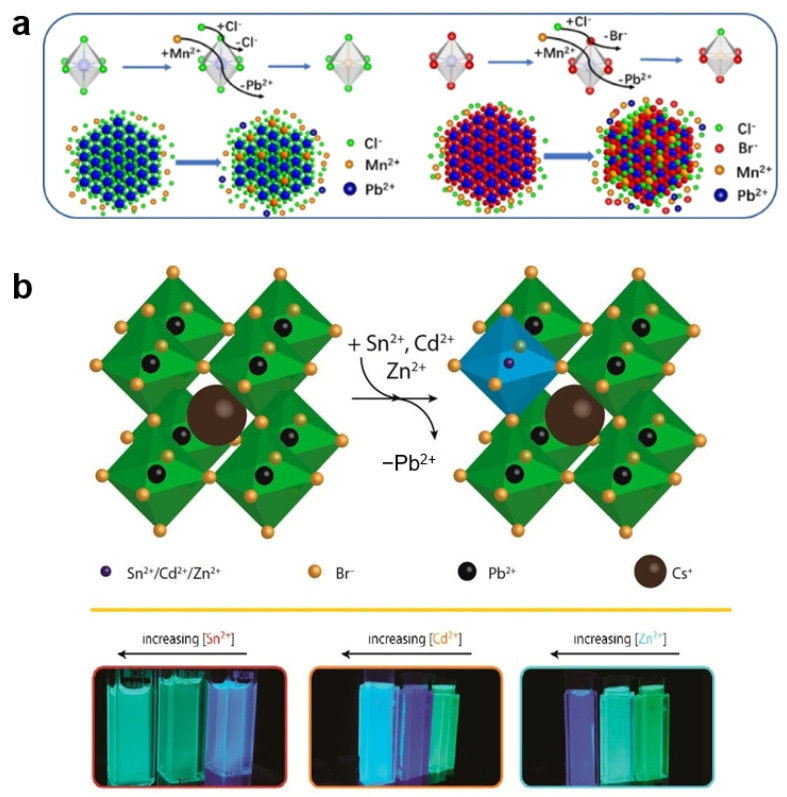
(**a**) Mn-to-Pb cation exchange assisted with dynamic halide exchange in CsPbX_3_ NCs [80]; (**b**) B-site cation exchange between Pb^2+^ and other divalent metal cations of Sn^2+^, Cd^2+^, and Zn^2+^ [81].

**Figure 8 nanomaterials-15-00375-f008:**
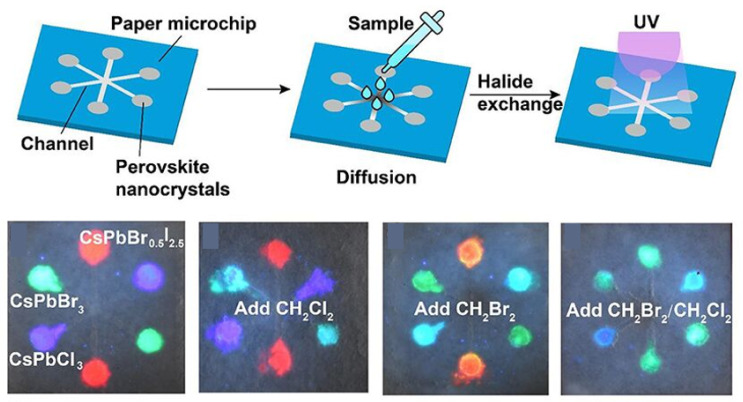
Schematic diagram and photos under UV light of a perovskite-based microfluidic sensor for on-site detection of haloalkanes [88].

**Figure 9 nanomaterials-15-00375-f009:**
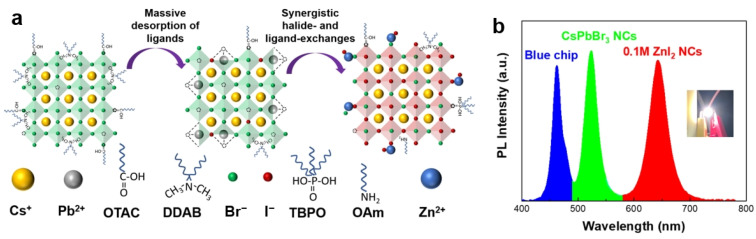
(**a**) Schematic diagram of the entire exchange process from the green CsPbBr_3_ NCs to the red CsPbBr_x_I_3−x_ NCs; (**b**) PL spectra of the white LED and a photograph of an operating device [22].

**Figure 10 nanomaterials-15-00375-f010:**
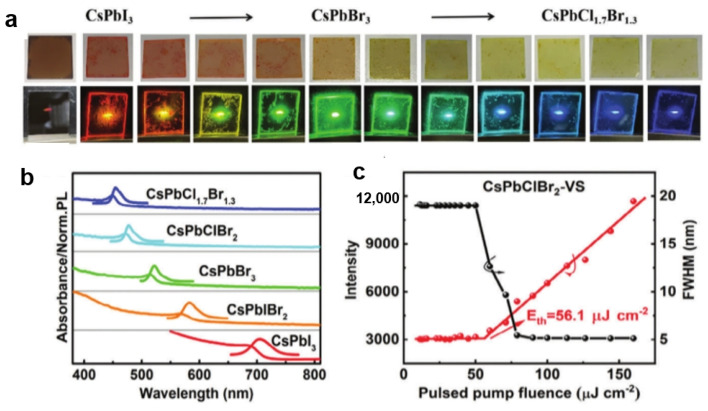
(**a**) Photographs of the as-prepared CsPbCl_x_Br_3−x_ and CsPbI_3−x_Br_x_ films unpumped and pumped by laser; (**b**) the absorption and PL spectra of a series of prepared CsPbX_3_ films; (**c**) the ASE threshold and FWHM behaviors of the prepared CsPbClBr_2_ films [94].

**Figure 11 nanomaterials-15-00375-f011:**
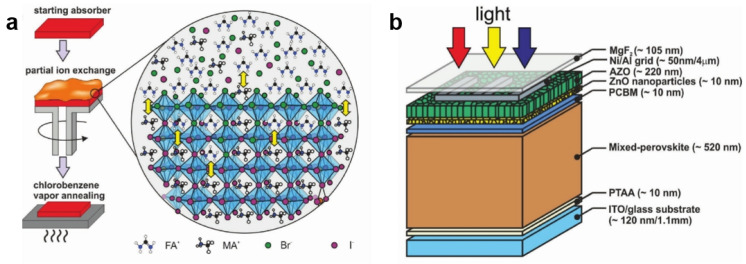
(**a**) Cartoon representation of the process of forming a gradient-composition perovskite layer through partial anion exchange; (**b**) schematic device structure of transparent perovskite solar cells [97].

## Data Availability

No new data were created or analyzed in this study.

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
