# Peer review of "Research Advances in Ion Exchange of Halide Perovskites"

_nanomaterials, 2025, doi:10.3390/nano15050375_

Round 1

Reviewer 1 Report

Comments and Suggestions for Authors  

The article is interesting. It's a pity that it is only a review, but it is still worth publishing because it provides a well-structured and comprehensive summary of recent advancements in ion exchange of halide perovskites. The review effectively covers the mechanisms, strategies, and applications of ion exchange, making it a valuable resource for researchers. While it does not introduce groundbreaking experimental results, it presents a clear and concise synthesis of existing knowledge, highlights key challenges, and suggests directions for future research. This makes it a relevant and informative contribution to the field.

Reviewer 2 Report

Comments and Suggestions for Authors

Reviewer 3 Report

Comments and Suggestions for Authors

This review article presents an overview on the mechanisms governing the ion exchange in halide perovskites. Moreover, the applications of ion exchanged perovskites in microfluidic sensors, light-emitting diodes (LEDs), lasers and solar cells are presented.

The topic is interesting and new right now. However, the manuscript needs to be revised and the following main points need to be carefully addressed before it can be considered for publication:

  1. Authors should clarify in detail in the manuscript what is the added value of this review article among the vast existing literature on the topic.
  2. In the introduction, before focusing on perovskite materials, the discussion should be extended to alternative systems which compete with perovskites for the same applications as solar cells, LEDs, laser and so on. Therefore, together with the quantum dots (0D) of different materials reported in ref [1], it should be briefly reported on challenging engineered architectures based on semiconductor nanostructures (1D), and on low-dimensional systems (2D) such as the ones studied in these relevant papers [https://doi.org/10.1016/j.jcrysgro.2006.10.107; https://doi.org/10.1063/1.3578189; https://doi.org/10.1063/1.111592] which are worth mentioning. This will allow to compare the benefits and performances of different systems, justifying the choice to focus on perovskite materials, for better understand the importance of this article.
  3. The two mechanisms of diffusion and exchange are reported in line 75, after which the diffusion mechanism is analyzed, which however is not described clearly enough.
  4. An important aspect of the studied structures is related to their preparation methods. Therefore, at least a short paragraph on their synthesis could be inserted along with the chemical reactions involved. Then, it is important to mention how much control and reproducibility can be achieved through the preparation method in the realization of such structures.
  5. At line 156 and following, the work of Xu et al. is mentioned. It would be useful if the emission spectrum mentioned by Xu Ref. [38] was reported and commented within the manuscript.
  6. It is well known that one of the limitations of perovskite materials is stability over time. The authors should carefully comment somewhere in the manuscript on this crucial point.
  7. Figure 11 seems to be poorly commented on; therefore, it should be deeply studied and critically commented on and then compared with the rest of the literature.
  8. The conclusions should clearly show the added value of this work in the field, should better identify the advantages but also the limitations of the presented results and outline possible future developments/perspectives.
  9. It seems that Ref [17] reports a text that is in Chinese language, therefore not understandable to a wide spectrum of users and should be, therefore, replaced with a more accessible one.
  10. All the references need to be completed with DOI.

Round 2

Reviewer 3 Report

Comments and Suggestions for Authors

The authors have done a good job by correctly and accurately addressing all the issues raised and have thus significantly improved the article. However, there are still the following issues that needs to be corrected:

  1. Ref [1]: the article number, which is 074312, is missing and should be added after the volume number;
  2. Ref [4]: the article number, which is 153106, is missing and should be added after the volume number;
  3. Ref [11]: the article number, which is 2400421, is missing and should be added after the volume number.

It is better to check all the references to be sure that they are correctly reported. Once the abovementioned issues will correctly be addressed, the manuscript will be ready for publication in the journal.
